# Properties of Hot Nuclear Matter

**Omar Benhar [1,*]** [ID]**, Alessandro Lovato [2,3] and Lucas Tonetto [4]**

1 INFN, Sezione di Roma, 00185 Roma, Italy
2 Physics Division, Argonne National Laboratory, Argonne, IL 60439, USA; lovato@alcf.anl.gov
3 INFN, Trento Institute of Fundamental Physics and Applications, 38123 Trento, Italy
4 Dipartimento di Fisica, Sapienza Università di Roma, 00185 Roma, Italy; lucas.tonetto@uniroma1.it
* Correspondence: omar.benhar@roma1.infn.it

**Abstract:** A fully quantitative description of the equilibrium and dynamical properties of hot nuclear matter will be needed for the interpretation of the available and forthcoming astrophysical data, providing information on the post-merger phase of a neutron star coalescence. We discuss the results of a recently developed theoretical model, based on a phenomenological nuclear Hamiltonian including two- and three-nucleon potentials, to study the temperature dependence of average and single-particle properties of nuclear matter relevant to astrophysical applications. The potential of the proposed approach for describing dissipative processes leading to the appearance of bulk viscosity in neutron star matter is also outlined.

**Keywords:** nuclear matter; thermal effects; bulk viscosity; neutron stars

## 1. Introduction

The interpretation of the presently available and future astronomical data providing information on the post-merger phase of coalescing binary neutron stars will require an accurate description of the properties of dense nuclear matter at temperatures as high as 100 MeV [1–5]. Of great importance, in this context, will be the development of a consistent framework suitable for modelling both the equilibrium configurations—determining the equation of state (EOS) of neutron star matter—and dissipative processes, involving mechanisms that lead to the appearance of bulk viscosity [6] and neutrino emission [7].

The EOS of hot nuclear matter is often derived from dynamical models based on the independent-particle approximation, using Skyrme-type effective interactions [8] or the formalism of quantum field theory and the relativistic mean field (RMF) approximation [9]. More comprehensive studies have been performed within the framework of Nuclear Many-Body Theory, in which the description of nuclear dynamics is based on a phenomenological Hamiltonian strongly constrained by the observed properties of the two- and three-nucleon systems. Calculations along this line have been carried out using both *G*-matrix perturbation theory and the variational approach based on the formalism of correlated wave functions and the cluster expansion technique; see, e.g., Refs. [10,11].

The authors of Ref. [12] have developed a procedure to renormalise the coordinate-space nuclear Hamiltonian by introducing screening effects arising from short-range nucleon–nucleon correlations. The resulting density-dependent effective potential—which includes the contributions of both two- and three-nucleon forces—is well behaved and can be employed to carry out perturbative calculations on the basis of the eigenstates of the noninteracting system. The extension of this formalism to the case of a nonzero temperature—involving a proper definition of the grand canonical potential needed to achieve thermodynamic consistency—is based on the assumption that at temperature $T \ll m_\pi$, $m_\pi \approx 138$ MeV, which is the isospin averaged mass of the $\pi$-meson, thermal effects do not significantly affect strong interaction dynamics [13].

In this paper, we discuss the main features of the approach of Refs. [12,13], as well as its application to a variety of equilibrium and dynamical properties of nuclear matter [14]. The

nuclear Hamiltonian and the derivation of the effective interaction are described in Section 2, while the perturbative calculation of the nuclear matter EOS at a finite temperature is outlined in Section 3. Thermal effects on the single-particle properties of charge-neutral $\beta$-stable matter are discussed in Section 5. Finally, Section 6 is devoted to the calculation of the bulk viscosity coefficient, driving the damping of the density oscillations of neutron stars.

## 2. Nuclear Hamiltonian

Nuclear Many-Body Theory (NMBT) is based on the hypothesis that all nucleon systems—from the deuteron to neutron stars—can be described in terms of point-like protons and neutrons, the dynamics of which are dictated by the Hamiltonian

$$H = \sum_i \frac{p_i^2}{2m} + \sum_{i<j} v_{ij} + \sum_{i<j<k} V_{ijk} \, , \tag{1}$$

with $m$ and $p_i$ being the mass and momentum of the $i$-th particle[1].

Nucleon–nucleon (NN) potentials that are local or semi-local in coordinate space are usually written in the form

$$v_{ij} = \sum_p v^p(r_{ij}) O_{ij}^p \, , \tag{2}$$

where $r_{ij} = |\mathbf{r}_i - \mathbf{r}_j|$ is the distance between the interacting particles. They are designed to reproduce the measured properties of the two-nucleon system, in both bound and scattering states, and reduce to the Yukawa one-pion-exchange potential at large distances. The sum in Equation (2) includes up to eighteen terms, with the corresponding operators, $O^p$, being required to describe the strong spin–isospin dependence and noncentral nature of nuclear forces ($i = 1, \ldots, 6$), as well as the occurrence of the spin–orbit and other angular-momentum-dependent interactions ($i = 7, \ldots 14$). Highly accurate phenomenological potentials, such as the Argonne $v_{18}$ (AV18) model, also feature additional terms accounting for small violations of charge symmetry and charge independence ($i = 15, \ldots, 18$) [15].

The addition of the three-nucleon (NNN) potential $V_{ijk}$ is needed to model the effects of *irreducible* three-body interactions, reflecting the appearance of processes involving the internal structure of the nucleons. Nuclear Hamiltonians comprising an AV18 NN potential and a phenomenological NNN potential, designed to explain the binding energies of $^3$He and $^4$He and the empirical equilibrium density of isospin-symmetric matter—such as the widely used Urbana IX (UIX) model [16,17]—have been shown to possess a remarkable predictive power. The results of Quantum Monte Carlo (QMC) calculations, extensively reviewed in Ref. [18], demonstrate that the AV18 + UIX Hamiltonian is capable of describing the energies of the ground and low-lying excited states of nuclei with mass number $A \leq 8$ with an accuracy within a few percent.

### 2.1. Renormalisation of the Nucleon–Nucleon Interaction

Owing to the presence of a strong repulsive core, the matrix elements of the NN potential between eigenstates of the noninteracting system are large, and standard perturbation theory cannot be used to carry out calculations of nuclear-matter properties.

The renormalisation scheme based on the formalism of correlated basis functions (CBFs) and the cluster expansion technique [19,20] allows one to determine an effective interaction suitable to carry out perturbative calculations in coordinate space. This approach, originally proposed by Cowell and Pandharipande in the early 2000s [21,22] and further developed by the authors of Refs. [23–25], has been extensively employed to study nuclear matter and neutron stars [12–14]. The resulting potential can be written as in Equation (2), including terms with $i \leq 6$, associated with the operators

$$O_{ij}^{p\leq6} = [1, (\sigma_i \cdot \sigma_j), S_{ij}] \otimes [1, (\tau_i \cdot \tau_j)] \, . \tag{3}$$

Here, $\sigma_i$ and $\tau_i$ are Pauli matrices acting in spin and isospin space, respectively, while the angular dependence is described by the tensor operator $S_{ij}$, defined as

$$S_{ij} = \frac{3}{r_{ij}^2}(\sigma_i \cdot \mathbf{r}_{ij})(\sigma_j \cdot \mathbf{r}_{ij}) - (\sigma_i \cdot \sigma_j) \, . \tag{4}$$

Note that the one-pion-exchange potential can also be written in terms of the $O_{ij}^{p\leq 6}$ defined by Equations (3) and (4).

The CBF effective interaction is *defined* by the equation

$$\langle H \rangle = \langle \Psi_0 | H | \Psi_0 \rangle = T_F + \langle \Phi_0 | \sum_{i<j} v_{ij}^{\text{eff}} | \Phi_0 \rangle \, , \tag{5}$$

where $|\Phi_0\rangle$ and $T_F$ denote the ground state of the noninteracting Fermi gas at density $\varrho$ and the corresponding energy, respectively, while $H$ is the nuclear Hamiltonian of Equation (1). The *correlated* ground state, $|\Psi_0\rangle$, is obtained from the corresponding Fermi gas state $|\Phi_0\rangle$ through the transformation

$$|\Psi_0\rangle \equiv \frac{F|\Phi_0\rangle}{\langle \Phi_0 | F^\dagger F | \Phi_0 \rangle^{1/2}} \, , \tag{6}$$

where the operator $F$ is a symmetrised product of two-body correlation operators, whose structure is chosen in such a way as to reflect the complexity of NN forces.

The effective interaction employed to obtain the results discussed in this paper has been derived following the procedure described in Ref. [24], which allows for the inclusion of the contribution of three-nucleon clusters to the ground-state expectation value $\langle H \rangle$ appearing in the left-hand side of Equation (5). This feature is essential to take into account three-nucleon interactions, which play a dominant role in the high-density regime relevant to astrophysical applications. It should be also pointed out that the nucleon–nucleon correlation functions are determined in such a way as to simultaneously reproduce the ground-state energies of pure neutron matter (PNM) and isospin-symmetric matter (SNM) obtained from highly accurate many-body calculations, carried out using the Auxiliary Field Diffusion Monte Carlo (AFDMC) technique or the Fermi Hyper-Netted Chain/Single-Operator Chain (FHNC/SOC) [12] variational approach. Therefore, the potential $v_{ij}^{\text{eff}}$ defined by Equation (5) can be used to describe nuclear dynamics in nuclear matter with any neutron excess.

### 2.2. The CBF Effective Interaction

The nuclear Hamiltonian employed to obtain the CBF effective interaction consists of the Argonne $v_6'$ (AV6P) NN potential—determined by projecting the full AV18 potential onto the operator basis of Equation (3) [26]—and the UIX NNN potential [17]. The AV6P potential predicts the binding energy and electric quadrupole moment of the deuteron with accuracies of 1%, and 4%, respectively, and provides an excellent fit of the elastic NN scattering phase shifts in the $^1S_0$ channel—which is dominant in pure neutron matter—up to lab energy $\sim$600 MeV, well above the pion production threshold.

The UIX potential is written in the form

$$V_{ijk} = V_{ijk}^{2\pi} + V_{ijk}^R \, , \tag{7}$$

where the first term is the attractive Fujita–Miyazawa potential [27]—describing two-pion-exchange NNN interactions with excitation of a $\Delta$-resonance in the intermediate state—while the purely phenomenological $V_{ijk}^R$ is an isoscalar repulsive term, the strength of which is fixed in such a way as to reproduce the saturation density of isospin-symmetric matter inferred from nuclear systematics [16,17].

Note that the CBF effective interaction depends on density through both dynamical correlations, described by the operator $F$ of Equation (6), and statistical correlations, arising

from the antisymmetric nature of the state $|\Phi_0\rangle$. In Figure 1, the radial dependence of $v^{\text{eff}}$ in the spin–isospin channel $S = 0$ and $T = 1$ is displayed for baryon densities $\varrho = 0.04$, 0.32, and 0.48 fm$^{-3}$. The effect of renormalisation clearly emerges from the comparison with the bare AV6P potential.

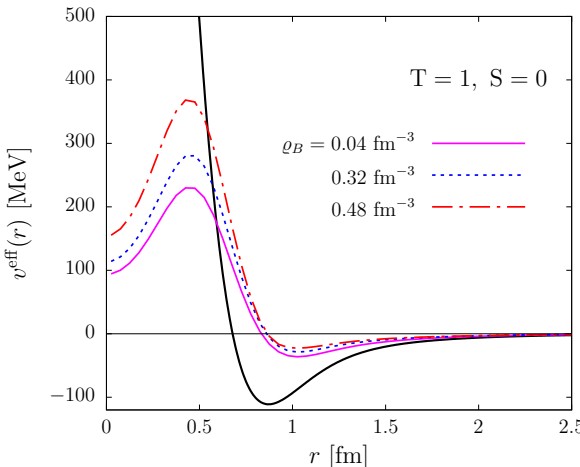

**Figure 1.** Radial dependence of the CBF effective potential in the spin–isospin $S = 0$, $T = 1$ channel. The solid, dashed, and dot-dash lines correspond to baryon number densities $\varrho = 0.04$, 0.32, and 0.48 fm$^{-3}$. For comparison, the thick solid line shows the bare AV6P potential.

Recent studies of the EOS of cold neutron matter—performed by the authors of Ref. [28] using accurate computational techniques—show that the predictions of the somewhat simplified AV6P + UIX Hamiltonian are very close to those obtained from the full AV18 + UIX model employed by Akmal, Pandharipande, and Ravenhall [29].

### 3. Many-Body Perturbation Theory at Finite Temperature

Let us consider, for simplicity, a one-component Fermi system. The derivation of perturbation theory at a finite temperature is based on the solution of the Bloch equation

$$-\frac{\partial \Phi}{\partial \beta} = (H - \mu N)\Phi \,, \tag{8}$$

where

$$\Phi(\beta) = e^{-\beta(H - \mu N)} \,, \tag{9}$$

with the initial condition $\Phi(0) = 1$; see, e.g., Ref. [30]. In the above equations, $\beta = 1/T$, while $H$ and $\mu$ denote the Hamiltonian and the chemical potential, respectively.

The perturbative expansion of the grand canonical partition function $Z = \text{Tr}\,\Phi$ is easily obtained by exploiting the formal similarity between Equation (8) and the time-dependent Schrödinger equation of quantum mechanics and rewriting the Hamiltonian in the form

$$H = H_0 + H_I. \tag{10}$$

The substitution of Equation (10) into the right-hand side of the Bloch equation, leading to

$$-\frac{\partial \Phi}{\partial \beta} = [(H_0 - \mu N) + H_I]\Phi = (H_0' + H_I)\Phi \,,$$

shows that the formalism of time-dependent perturbation theory can be readily generalised by replacing $t \to -i\beta$ and using $H_0'$ to define operators in the interaction picture.

The fundamental relation

$$\Omega = -\frac{1}{\beta} \ln Z = -PV = F - \mu N = E - TS - \mu N \, , \tag{11}$$

where $V$ is the normalisation volume, provides the link between the grand canonical potential $\Omega$, the pressure $P$, and the free energy $F = E - TS$, with $E$ and $S$ being the energy and entropy of the system, respectively; see, e.g., Ref. [31]. From Equation (11), it follows that

$$P = -\frac{\Omega}{V} \quad , \quad S = -\frac{\partial \Omega}{\partial T} \quad , \quad N = -\frac{\partial \Omega}{\partial \mu} \, . \tag{12}$$

Here, we will discuss the application of the above results to a system described by the Hamiltonian (10), with

$$H_0 = \sum_k e_k a_k^\dagger a_k \, , \tag{13}$$

where, in general

$$e_k = \frac{\mathbf{k}^2}{2m} + U_k = t_k + U_k \, , \tag{14}$$

and

$$H_I = \sum_{k,k',q,q'} \langle k'q' | v | kq \rangle a_{k'}^\dagger a_{q'}^\dagger a_q a_k - \sum_k U_k a_k^\dagger a_k \, . \tag{15}$$

Here, the label $k$ specifies both the particle momentum and the discrete quantum numbers corresponding to one-particle states, $a_k^\dagger$ and $a_k$ denote creation and annihilation operators, respectively, and $v$ is the potential describing interparticle interactions. The single-particle potential $U_k$, which in principle does not affect the results of calculations of physical quantities, is chosen in such a way as to improve the convergence of the perturbative expansion or fulfil specific conditions; see, e.g., Ref. [32].

It has to be pointed out that, according to Equation (11), the pressure can be written in the form

$$P = \varrho \left( \mu - \frac{F}{N} \right) \, , \tag{16}$$

with $\varrho = N/V$, implying that at equilibrium, that is, for $P = 0$, $\mu = F/N$. This result is the generalisation of the Hugenholtz–Van Hove theorem [33] to the case of a nonvanishing temperature.

It should be emphasised that, when used in conjunction with the CBF effective interaction discussed in Section 2.2, the perturbative approach based on the Hamiltonian defined by Equations (10) and (13)–(15) allows two- and three-nucleon interactions to be taken into account in a fully consistent fashion.

### 3.1. Perturbative Expansion

At the first order in $H_I$, the grand canonical potential is given by [34]

$$\Omega = \Omega_0 + \Omega_1 \, , \tag{17}$$

with

$$\Omega_0 = -\frac{1}{\beta} \sum_k \ln \left\{ 1 + e^{-[\beta(e_k - \mu)]} \right\} \, , \tag{18}$$

$$\Omega_1 = \frac{1}{2} \sum_{kk'} \langle kk'|v|kk'\rangle_A \, n_k n_{k'} - \sum_k U_k n_k \,, \tag{19}$$

where $|kk'\rangle_A = |kk'\rangle - |k'k\rangle$ denotes an antisymmetrised two-particle state, and $n_k$ is the Fermi distribution, defined as

$$n_k = \left[ 1 + e^{\beta(e_k - \mu)} \right]^{-1} . \tag{20}$$

From Equations (18) and (19) it follows that the free energy per particle,

$$\frac{F}{N} = \frac{1}{N}(\Omega_0 + \Omega_1) + \mu \,, \tag{21}$$

can be cast in the form

$$\frac{F}{N} = \frac{1}{N} \Bigg\{ \sum_k t_k n_k + \frac{1}{2} \sum_{k,k'} \langle kk'|v|kk'\rangle_A \, n_k n_{k'} \tag{22}$$
$$+ \frac{1}{\beta} \sum_k \left[ n_k \ln n_k + (1 - n_k) \ln(1 - n_k) \right] \Bigg\} + \mu \left( 1 - \frac{1}{N} \sum_k n_k \right) .$$

In principle, for any assigned values of temperature and chemical potential, the above equations provide a scheme for the determination of the equation of state of nuclear matter at a finite temperature, $P = P(\mu, T)$. However, because the baryon number is conserved by all known interactions, in nuclear matter it is more convenient to use the baryon density as an independent variable and determine the chemical potential from the relation

$$\varrho = -\frac{1}{V} \frac{\partial}{\partial \mu} (\Omega_0 + \Omega_1) . \tag{23}$$

In the $T \to 0$ limit, the momentum distribution reduces to the Heaviside step function $\theta(\mu - e_k)$, with the chemical potential being given by $\mu = e_{k_F}$. The Fermi momentum is related to the particle number density $\varrho$ through $k_F = \left( 6\pi^2 \varrho / \nu \right)^{1/3}$, where $\nu$ denotes the degeneracy of momentum eigenstates.

### 3.2. Thermodynamic Consistency

For $T \neq 0$ and density-dependent potentials, thermodynamic consistency is not trivially achieved at any given order of perturbation theory. A clear manifestation of this difficulty is the mismatch between the value of pressure obtained from Equation (16) and the one resulting from the alternative—although, in principle, equivalent—thermodynamic expression

$$P = -\frac{\partial F}{\partial V} = \varrho^2 \frac{\partial}{\partial \varrho} \frac{F}{N} . \tag{24}$$

A procedure fulfilling the requirement of thermodynamic consistency by construction can be derived from a variational approach, based on the minimisation of the trial grand canonical potential [35]

$$\widetilde{\Omega} = \sum_k t_k n_k + \frac{1}{2} \sum_{k,k'} \langle kk'|v|kk'\rangle_A \, n_k n_{k'} \tag{25}$$
$$+ \frac{1}{\beta} \sum_k \left[ n_k \ln n_k + (1 - n_k) \ln(1 - n_k) \right] - \mu N,$$

with respect to the form of distribution $n_k$. Note that the above expression—the use of which is fully legitimate in the variational context—can also be obtained in first-order perturbation theory neglecting terms involving $\partial \Omega_1 / \partial T$ and $\partial \Omega_1 / \partial \mu$ [34].

The condition,

$$\frac{\delta \widetilde{\Omega}}{\delta n_k} = 0 , \tag{26}$$

turns out to be satisfied by the distribution function

$$n_k = \left\{ 1 + e^{\beta[(t_k + U_k + \delta e) - \mu]} \right\}^{-1} , \tag{27}$$

with

$$U_k = \sum_{k'} \langle kk' | v | kk' \rangle_A \, n_{k'} , \tag{28}$$

and

$$\delta e = \frac{1}{2} \sum_{k,k'} \langle kk' | \frac{\partial v}{\partial \varrho} | kk' \rangle_A \, n_k n_{k'} . \tag{29}$$

Within the above scheme, which reduces to the standard Hartee–Fock approximation in the case of density-independent potentials, all thermodynamic functions at a given temperature and baryon density can be consistently obtained using the distribution $n_k$ of Equation (27). Note, however, that, because both $U_k$ and $\delta e$ depend on $n_k$ (see Equations (28) and (29)), calculations must be carried out self-consistently, applying an iterative procedure.

## 4. Equilibrium Properties of Hot Nuclear Matter

Consider now a nucleon system at temperature $T$, baryon number density, $\varrho$ and proton number density $\varrho_p = Y_p \, \varrho$. At the first order in the CBF effective interaction $v^{\text{eff}}$, the internal energy per nucleon can be written in the form [14]

$$\frac{E}{N} = \frac{1}{N} \left\{ \sum_{\alpha \mathbf{k}} \frac{\mathbf{k}^2}{2m} \, n_\alpha(k, T) + \frac{1}{2} \sum_{\alpha \mathbf{k}} \sum_{\alpha' \mathbf{k}'} \langle \alpha k, \alpha' k' | v^{\text{eff}} | \alpha k, \alpha' k' \rangle_A \, n_\alpha(k, T) n_{\alpha'}(k', T) \right\} . \tag{30}$$

In the above equations, the index $\alpha = n, p$ labels neutrons and protons, respectively, $\mathbf{k}$ is the nucleon momentum, $k = |\mathbf{k}|$, and $|\alpha k, \alpha' k' \rangle_A$ denotes an antisymmetrised two-nucleon state. The conservation of the baryon number obviously implies that $Y_n = 1 - Y_p$.

The temperature dependence is described by the Fermi distributions

$$n_\alpha(k, T) = \left\{ 1 + \exp \left[ \beta(e_{\alpha k} - \mu_\alpha) \right] \right\}^{-1} , \tag{31}$$

where the single-particle energies are defined as

$$e_{\alpha k} = e_{\alpha k}^{\text{HF}} + \delta e , \tag{32}$$

with

$$e_{\alpha k}^{\text{HF}} = \frac{\mathbf{k}^2}{2m} + \sum_{\alpha' \mathbf{k}'} \langle \alpha k, \alpha' k' | v^{\text{eff}} | \alpha k, \alpha' k' \rangle_A \, n_\alpha(k', T) , \tag{33}$$

and

$$\delta e = \frac{\varrho}{2} \sum_{\alpha \mathbf{k}} \sum_{\alpha' \mathbf{k}'} \langle \alpha k, \alpha' k' | \frac{\partial v^{\text{eff}}}{\partial \varrho} | \alpha k, \alpha' k' \rangle_A \, n_\alpha(k, T) n_{\alpha'}(k', T), \tag{34}$$

The correction to the Hartree–Fock (HF) spectrum is needed to satisfy the requirement of thermodynamic consistency and vanishes in the case of a density-independent potential [13]. The chemical potentials $\mu_\alpha$ are determined by the normalisation conditions

$$2 \sum_{\alpha \mathbf{k}} n_\alpha(k, T) = N_\alpha \, , \tag{35}$$

where $N_\alpha$ denotes the number of particles of species $\alpha$, the fractional density of which is $Y_\alpha = N_\alpha / N_B = \varrho_\alpha / \varrho$. Note that the above definitions imply that both the single-nucleon energies and the chemical potentials depend on temperature through the Fermi distribution.

The free energy per nucleon is obtained from

$$\frac{F}{N} = \frac{1}{N}(E - TS) \, , \tag{36}$$

with the internal energy per nucleon of Equation (30) and the entropy per nucleon defined as

$$\frac{S}{N} = -\sum_{\alpha \mathbf{k}} \left\{ n_\alpha(k, T) \ln n_\alpha(k, T) + [1 - n_\alpha(k, T)] \ln [1 - n_\alpha(k, T)] \right\} . \tag{37}$$

Figure 2 shows the density and temperature dependence of the free energy per nucleon of SNM and PNM, corresponding to proton fraction $Y_p = 0.5$ and 0, respectively, obtained from the procedure described above using the CBF effective interaction.

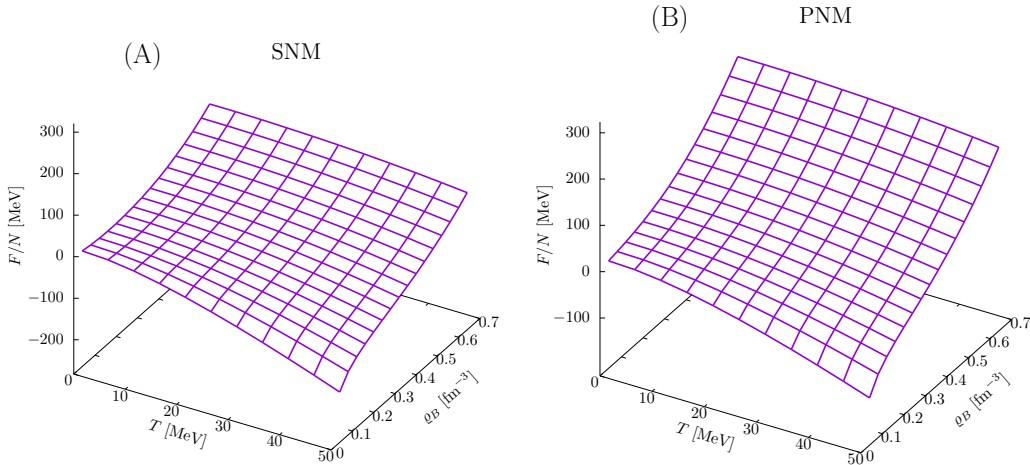

**Figure 2.** Density and temperature dependence of the free energy per nucleon of SNM (**A**) and PNM (**B**), computed using Equations (30)–(37), with the CBF effective interaction discussed in Section 2.2.

## 5. Thermal Effects on Nuclear Matter Properties

In the temperature regime discussed in this paper, thermal modifications of nuclear matter properties arise primarily from the Fermi distributions, defined by Equation (31). Comparison to the $T \to 0$ limit shows that the probability distribution $n_\alpha(k, T > 0)$ is reduced from unity in the region corresponding to $\mu_\alpha - T \lesssim e_{\alpha k} \lesssim \mu_\alpha$ and acquires nonvanishing positive values for $\mu_\alpha \lesssim e_{\alpha k} \lesssim \mu_\alpha + T$. It follows that, for any given temperature $T$, the extent of thermal modifications to the Fermi distribution is driven by the ratio $2T/\mu_\alpha$. This observation in turn implies that, because the chemical potential is a monotonically increasing function of the particle density $\varrho_\alpha$ over a broad range of temperatures, for any given $T$ thermal effects turn out to be more significant at lower $\varrho_\alpha$. However, they become vanishingly small in the high-density regime, in which degeneracy dominates. For example, in SNM the nucleon chemical potential grows from 32 to 144 MeV as the baryon number density increases from 0.32 to 0.48 fm$^{-3}$ [13].

*Charge-Neutral β-Stable Matter at Finite Temperature*

In charge-neutral matter consisting of neutrons, protons, and leptons in equilibrium with respect to the weak interaction processes

$$n \to p + \ell + \bar{\nu}_\ell \quad , \quad p + \ell^- \to n + \nu_\ell \,, \tag{38}$$

where $\ell = e, \mu$ labels the lepton flavour, the proton fraction $Y_p$ is uniquely determined by the equations

$$\mu_n - \mu_p = \mu_\ell \quad , \quad Y_p = \sum_\ell Y_\ell \,. \tag{39}$$

At densities such that the electron chemical potential does not exceed the rest mass of the muon, $m_\mu = 105.7$ MeV, the sum appearing in the above equation includes electrons only. At higher densities—typically at $\varrho \gtrsim \varrho_0$, with $\varrho_0 = 0.16$ fm$^{-3}$ being the baryon number density of isospin-symmetric matter in thermodynamic equilibrium at $T = 0$—the appearance of muons becomes energetically favoured and should be taken into account.

The solid lines of Figure 3 show the density dependence of the proton fractions corresponding to the $\beta$-equilibrium in matter consisting of protons, neutrons, electrons, and muons, or $npe\mu$ matter, at $T = 0$ (triangles) and 50 MeV (circles) [14]. The results have been obtained assuming that neutrinos do not interact with matter and therefore have vanishing chemical potential. For comparison, the same quantities in $npe$ matter, in which the muon contribution is not included, are displayed by the dashed lines. At temperatures as high as 50 MeV, the assumption that neutrinos are nondegenerate may be questionable, because the neutrino mean free path in matter, $\lambda$, is known to decrease with increasing temperatures. The present analysis should be seen as the first step in a systematic study of neutrino trapping, based on a fully consistent evaluation of $\lambda$, which is currently underway.

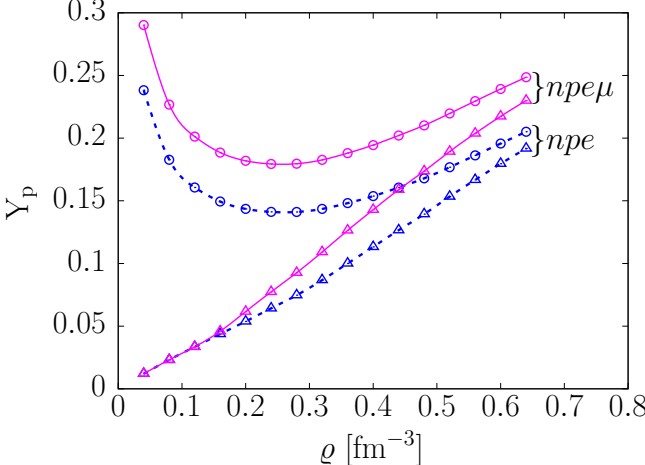

**Figure 3.** Density dependence of the proton fraction in charge-neutral $\beta$-stable matter. Solid lines marked with triangles and circles correspond to $npe\mu$ matter at $T = 0$ and 50 MeV, respectively. The same quantities in $npe$ matter are represented by dashed lines. From Ref. [14].

The most prominent thermal effect emerging from Figure 3 is a significant departure from the monotonic behaviour observed in cold matter. The emergence of a minimum in the density dependence of the proton fraction—which turns out to be largely unaffected by the appearance om muons—results from the balance between the thermal and degeneracy contributions to the chemical potentials appearing in Equation (39). For $T \gtrsim 20$ MeV and a low density, typically $\varrho \lesssim \varrho_0$, the thermal contribution—whose leading order term can be written in the form $\delta\mu_\alpha \propto T^2/\varrho_\alpha^{1/3}$—turns out to be much larger for protons than for neutrons, and the $\beta$-equilibrium requires large proton fractions. The results displayed in

Figure 3, showing that $Y_p$ does not exceed 25% for $\varrho_0/2 \leq \varrho \leq 4\varrho_0$, imply that thermal effects in $\beta$-stable matter mainly affect the proton distributions.

The description of single-particle dynamics in interacting many-body systems is largely based on the use of the effective mass $m_\alpha^\star$, defined as

$$\frac{1}{m_\alpha^\star} = \left( \frac{1}{k} \frac{de_{\alpha k}}{dk} \right)_{k=k_{F_\alpha}} , \tag{40}$$

with $e_{\alpha k}$ given by Equation (32). The effective mass dictates the nucleon dispersion relation in matter, which plays a critical role in determining the rates of many processes relevant to neutron star properties, such as the bulk viscosity to be discussed in Section 6. The momentum dependence of the nucleon spectra in cold nuclear matter is often parametrised according to [7]

$$e_{\alpha k} = \frac{k^2}{2m_0^\star} + U_\alpha , \tag{41}$$

where $m_0^\star$ denotes the value of $m_\alpha^\star$ at $T = 0$, while the offset $U_\alpha$ from the free-nucleon spectrum—the value of which depends on *both* the temperature and density—is determined by the requirement that the above approximation reproduce the results of the full microscopic calculation at $k = 0$.

The solid lines of Figure 4 show the momentum dependence of the proton spectra in charge-neutral $\beta$-stable matter at baryon density $\varrho = 2\varrho_0$ and temperature $T = 0$ and 50 MeV. A comparison with the dashed lines indicates that at $T = 50$ MeV the apprroximation of Equation (41) fails to provide an accurate representation of $e_{\alpha k}$ for $k > 0.5$ fm$^{-1}$. However, the quadratic approximation turns out to be in remarkable agreement with the microscopic result if $m_0^\star$ is replaced with the effective mass computed at $T = 50$ MeV. At $\varrho = 2\varrho_0$ ($3\varrho_0$), the values of $U_\alpha$ turn out to increase from $-98$ ($-50$) MeV to $-62$ ($-10$) MeV as the temperature grows from 0 to 50 MeV.

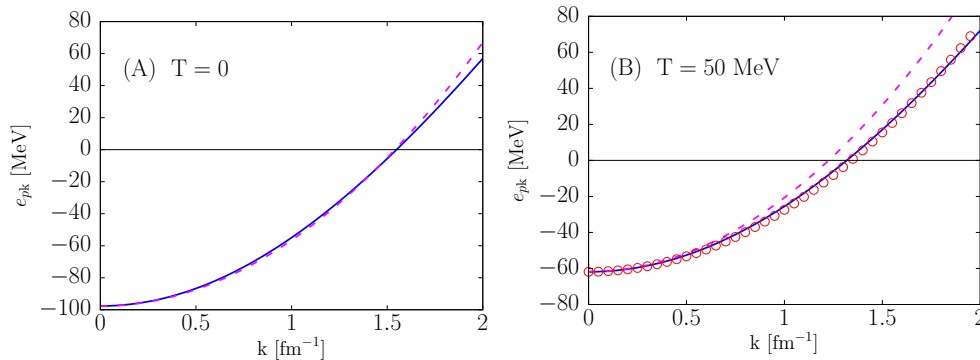

**Figure 4.** Momentum dependence of the proton spectrum in charge-neutral $\beta$-stable matter at baryon density $\varrho = 2\varrho_0$ and temperature $T = 0$ (**A**) and 50 MeV (**B**). The solid and dashed lines represent the results of full microscopic calculations and the approximation of Equation (41), respectively. The open circles in panel (**B**) have been obtained using the quadratic approximation with $m_0^\star$ replaced by the effective mass computed at $T = 50$ MeV.

## 6. Bulk Viscosity of Neutron Star Matter

Bulk viscosity in charge-neutral $\beta$-stable matter appears in the aftermath of an instantaneous departure from $\beta$-equilibrium, resulting from the change in density and pressure induced by some perturbation. The role of bulk viscosity in determining the maximum rotation rate of neutron stars—the value of which is limited by the onset of the Chandrasekahr–Friedman–Shutz (CFS) instability—driven by gravitational wave emission [36–38]—has been discussed by many authors; for a review, see, e.g., Ref. [39]. More recent studies, motivated by the observation of the gravitational wave event GW170817 [40] focused

on the effect of bulk viscosity in neutron star mergers, leading to a damping of density oscillations [41].

*6.1. Dissipative Processes in Fluids*

The description of fluids is based of the continuity equation expressing conservation of mass

$$\frac{\partial \varrho}{\partial t} = -\boldsymbol{\nabla} \cdot (\varrho \mathbf{v}) , \tag{42}$$

where $\mathbf{v} = \mathbf{v}(\mathbf{r}, t)$ and $\varrho = \varrho(\mathbf{r}, t)$ denote the velocity field and matter density, respectively. For ideal fluids, that is, in the absence of dissipation, the force acting on a fluid element is simply related to the pressure of the surrounding medium through

$$\mathbf{F} = -\boldsymbol{\nabla} P , \tag{43}$$

and the equation of motion reduces to Euler's equation [42]

$$\frac{\partial \mathbf{v}}{\partial t} + (\mathbf{v} \cdot \boldsymbol{\nabla}) \mathbf{v} = -\frac{1}{\varrho} \boldsymbol{\nabla} P . \tag{44}$$

The above equations can be combined to obtain

$$\frac{\partial (\varrho \mathrm{v}_i)}{\partial t} = -\nabla_j \Pi_{ij} , \tag{45}$$

where a sum on the index *j* is implicit and $\Pi_{ij}$ is the momentum flux tensor, defined as

$$\Pi_{ij} = P\delta_{ij} + \varrho \mathrm{v}_i \mathrm{v}_j . \tag{46}$$

In viscous fluids, the form of the continuity equation does not change, but the momentum flux tensor needs to be modified by adding a term describing the deviations from the ideal fluid behaviour, due the occurrence of processes involving *irreversible* momentum transfer. The resulting expression can be written in the form

$$\Pi_{ij} = P\delta_{ij} + \varrho \mathrm{v}_i \mathrm{v}_j + \delta \Pi_{ij} , \tag{47}$$

where

$$\delta\Pi_{ij} = -\eta \left[ \nabla_j \mathrm{v}_i + \nabla_i \mathrm{v}_j + \frac{2}{3}\delta_{ij}(\boldsymbol{\nabla} \cdot \mathbf{v}) \right] - \zeta\delta_{ij}(\boldsymbol{\nabla} \cdot \mathbf{v}) , \tag{48}$$

with the two velocity-independent quantities $\eta$ and $\zeta$ being referred to as shear and bulk viscosity coefficients, respectively. In the case of incompressible fluids, in which $\varrho$ does not depend on either $\mathbf{r}$ or $t$, Equation (42) implies

$$\boldsymbol{\nabla} \cdot \mathbf{v} = 0 , \tag{49}$$

and the contribution of bulk viscosity vanishes.

A pulsation of frequency $\omega$ induces a variation of the fluid density described by the equation

$$\varrho(t) = \varrho_{\mathrm{eq}} + \delta\varrho \cos \omega t , \tag{50}$$

where $\varrho_{\mathrm{eq}}$ is the density corresponding to chemical equilibrium and $(\delta\varrho/\varrho_{\mathrm{eq}}) \ll 1$. The rate of energy density dissipation due to bulk viscosity, averaged over the pulsation period $\tau = 2\pi/\omega$, is given by [43]

$$\left\langle \frac{d\epsilon_{\text{diss}}}{dt} \right\rangle = -\frac{1}{\tau} \int_0^\tau dt \, \nabla_i \left[ -v_j \, \zeta \delta_{ij} (\nabla \cdot \mathbf{v}) \right] = \frac{1}{\tau} \int_0^\tau dt \, \zeta (\nabla \cdot \mathbf{v})^2 \, . \tag{51}$$

*6.2. Bulk Viscosity of β-Stable Matter*

The authors of Refs. [44,45] have derived the expression of the bulk viscosity coefficient of neutrino-transparent β-stable *npe* matter under the assumption that neutrinos and antineutrinos are produced through the processes

$$p + e \to n + \nu_e \, , \tag{52}$$

$$n \to p + e + \bar{\nu}_e \, , \tag{53}$$

referred to as *direct* Urca reactions. They followed the scheme originally proposed in Ref. [46] and employed a simple parametrisation of the nuclear matter EOS assuming a quadratic dependence on the neutron excess $\alpha = 1 - 2Y_p$.

In direct Urca processes leading to the emission of nondegenerate neutrinos, the requirement of momentum conservation entails a relation involving the momenta of the degenerate fermions, which can be exploited to define a threshold by applying the *Fermi surface approximation*. While being admittedly inconsistent at $T > 0$—and certainly not expected to be accurate at $T = 50$ MeV—in our work this definition is used as a baseline to analyse thermal modifications of the onset of the Urca process originating from the modifications of matter composition illustrated in Figure 3.

A comprehensive and detailed analysys of the appearance of bulk viscosity associated with the occurrence of Urca processes in *npe* and *npeμ* matter—including both the neutrino-transparent and neutrino-trapped regimes—has recently been carried out by the authors of Ref. [47].

Bulk viscosity is associated with deviations from beta equilibrium, signalled by a nonvanishing difference between the neutrino and antineutrino production rates

$$\Delta\Gamma = \Gamma_{\nu_e} - \Gamma_{\bar{\nu}_e} \, , \tag{54}$$

with $\Delta\Gamma$ being a function of the variable $\delta\mu$, describing the departure from chemical equilibrium. In the case of nondegenerate neutrinos,

$$\delta\mu = \mu_n - \mu_p - \mu_e \, . \tag{55}$$

Under the assumption that $\delta\mu \ll T \ll \mu_i$, $\Delta\Gamma$ can be expanded in powers of $\delta_\mu$. At the leading order one finds

$$\Delta\Gamma = \lambda\delta_\mu \, , \tag{56}$$

with

$$\lambda = 2 \left( \frac{\partial\Gamma_\nu}{\partial\delta\mu} \right)_{\delta\mu=0} \, . \tag{57}$$

The energy density dissipation rate—averaged over the period of the density fluctuation of Equation (50)—can be written in terms of the change in pressure associated with the departure from chemical equilibrium using

$$\begin{aligned} \left\langle \frac{d\epsilon_{\text{diss}}}{dt} \right\rangle &= \frac{1}{\varrho} \frac{1}{\tau} \int_0^\tau dt \, \delta P_{\text{chem}} \frac{d\varrho}{dt} \\ &= \frac{\delta\varrho}{\varrho} \frac{\omega}{\tau} \int_0^\tau dt \, \delta P_{\text{chem}} \, \sin\omega t \, , \end{aligned} \tag{58}$$

with [46]

$$\delta P_{\text{chem}} = -\lambda C^2 \frac{\delta \varrho}{\varrho} \frac{\omega}{\omega^2 + (2\lambda B/\varrho)^2} \sin \omega t + \ldots \tag{59}$$

where the ellipsis refers to additional terms giving vanishing contributions to the integral. Here, $\lambda$ is given by Equation (57), and the constants $B$ and $C$ are defined as

$$B = \left(\frac{\partial \delta \mu}{\partial \alpha}\right)_{\delta \varrho = 0} \, , \quad C = \left[\varrho \left(\frac{\partial \delta \mu}{\partial \varrho}\right)\right]_{\alpha = \alpha_{\text{eq}}} , \tag{60}$$

where $\alpha_{\text{eq}}$ is the value of neutron excess corresponding to the $\beta$-equilibrium. The substitution of Equation (59) into Equation (58) yields

$$\left\langle \frac{d\epsilon_{\text{diss}}}{dt} \right\rangle = -\lambda \frac{\omega^2}{2} \left(\frac{\delta \varrho}{\varrho}\right)^2 \frac{C^2}{\omega^2 + (2B\lambda/\varrho)^2} \, .$$

The bulk viscosity coefficient is obtained combining the above equation with Equation (51), which can be rewritten using the continuity equation associated with the conservation of the baryon number, implying

$$\boldsymbol{\nabla} \cdot \mathbf{v} = -\frac{1}{\varrho} \frac{\partial \varrho}{\partial t} = \omega \frac{\delta \varrho}{\varrho} \sin \omega t \, . \tag{61}$$

Following this procedure, one finally arrives at

$$\zeta = -\lambda \frac{C^2}{\omega^2 + (2B\lambda/\varrho)^2} \, . \tag{62}$$

The above result shows that the calculation of the bulk viscosity coefficient involves $\lambda$, which is obtained from the neutrino and antineutrino production rates, and the constants $B$ and $C$ defined by Equation (60). We have studied the density and temperature dependence of $\zeta$ using the nuclear matter model described in the previous sections, which allows us to take into account thermal modifications of the chemical potentials and effective masses within a fully consistent framework.

*6.3. Calculation of the Bulk Viscosity Coefficient*

The bulk viscosity coefficient $\zeta$ has been obtained from Equation (62), with $B$ and $C$ given by Equation (60). The calculation of the coefficient $\lambda$, defined by Equations (56) and (57), involves the rate of production of antineutrinos of flavour $\ell$ in neutron decay processes, which can be written in the form

$$\Gamma_{\bar{\nu}_\ell} = \int \frac{d^3 p_n}{(2\pi)^3} \frac{d^3 p_p}{(2\pi)^3} \frac{d^3 p_\ell}{(2\pi)^3} \frac{d^3 p_\nu}{(2\pi)^3} f_n (1 - f_p)(1 - f_\ell)$$
$$\times (2\pi)^4 \delta(E_n - E_p - E_\ell - E_\nu)\delta^{(3)}(\boldsymbol{p}_n - \boldsymbol{p}_p - \boldsymbol{p}_\ell - \boldsymbol{p}_\nu) \sum_{\text{spins}} |M|^2 , \tag{63}$$

as well as the rate of neutrino production associated with lepton capture by protons. In the above equation, $\boldsymbol{p}_j$—with $j = n, p, \ell, \nu$—denotes the particle momentum, while the $f_j$ are the Fermi distributions of degenerate particles, defined by Equation (20). Treating the nucleons as nonrelativistic and neglecting $\boldsymbol{p}_\nu$ in the argument of the momentum-conserving $\delta$-function, as justified for thermal neutrinos, the spin-summed squared transition amplitude can be written in the form

$$\sum_{\text{spins}} |M|^2 = 2G^2(g_V^2 - 3g_A^2) \, . \tag{64}$$

Here, $G = G_F \cos\theta_c$, with $G_F$ and $\theta_c$ being the Fermi constant and the Cabibbo mixing angle, respectively, while $g_V = 1$ and $g_A = -1.26$ denote the vector and axial-vector coupling constants.

The above transition probability is a constant, which can be moved out of the integral appearing in the right-hand side of Equation (63). The integrations can then be performed either numerically, as described by the authors of Ref. [48], or applying the widely used approximation known as *phase space decomposition*, thoroughly discusssed in, e.g., Ref. [49]. Within this scheme, the magnitudes of the momenta of the strongly degenerate fermions—that is, the nucleons and the charged lepton—are replaced by the corresponding Fermi momenta, and the effective masses, defined as in Equation (40), are used to rewrite the integration measures in the form

$$d^3 p_j = d\Omega_j \, p_{Fj} m_j^\star \, dE_j \,, \tag{65}$$

where $j = n$, $p$, $\ell$, and $\Omega_j$ is the solid angle specifying the direction of the vector $\boldsymbol{p}_j$.

In this work, the calculation of the rates of neutrino and antineutrino production in the Urca processes (52) and (53) have been performed using an *improved* version of the *phase space decomposition*. The important new feature of this procedure lies in the inclusion of thermal modifications of the effective masses, which were found to be sizeable by the authors of Ref. [14]. Note that this entails a temperature dependence of the integration measures. Thermal modifications of the chemical potentials, determining both the Fermi distributions and matter composition, have also been taken into account.

Equation (62) can be written in a somewhat more transparent form in terms of the equilibration rate $\gamma = -2B\lambda/\varrho$. The resulting expression,

$$\zeta = \varrho \frac{C^2}{B} \frac{1}{2} \frac{\gamma}{\omega^2 + \gamma^2} \,, \tag{66}$$

exhibits a resonant maximum located at $\gamma = \omega$. This property, which depends on both the density and temperature, has been thoroughly analysed by the authors of Ref. [41], who also discussed the differences between the isothermal and adiabatic treatment of thermodynamic quantities.

Under the assumption that during a neutron star merger the heat flow between adjoining fluid elements is negligible, the derivatives involved in the calculations of the quantities $B$ and $C$ defined by Equation (60) must be computed keeping the entropy per baryon constant. The alternative procedure based on isothermal derivation, while being equivalent in the $T \to 0$ limit, leads to sizeably different results in the high-temperature regime.

Figure 5 shows the behaviour of $\zeta$ as a function of density for temperatures 10, 30, and 50 MeV, with the frequency of the density oscillation driving the appearance of viscosity being set to $\omega = 2\pi \times 1$ kHz, a value typical of oscillations occurring in neutron star mergers [6]. The results were obtained by performing the derivatives of Equation (60) using both the isothermal (solid lines and open circles) and adiabatic (dashed lines) definitions.

The peculiar density dependence featured by the solid line corresponding to the highest temperature, $T = 50$ MeV, results from the occurrence of a minimum in the proton fraction $Y_p(\varrho)$—clearly visible in Figure 3—and from the fact that the value of $Y_p$ exceeds the threshold for the onset of direct Urca processes at all densities. A comparison to the corresponding dashed line shows that the minimum of $\zeta(\varrho)$ is entirely washed out when the isothermal derivative is replaced by the adiabatic one. Because the proton fraction is still above the Urca threshold, however, one finds $\zeta(\varrho) \neq 0$ over the whole density range of the figure.

A similar pattern is displayed by the solid line and the open circles representing the results obtained at $T = 30$ MeV, although in this case the minimum of $Y_p(\varrho)$ is not reflected by a minimum in $\zeta(\varrho)$. One only observes an isolated point with $\zeta \neq 0$ located at $\rho = 0.25\,\rho_0$ and a density region extending up to $\rho > 2.25\,\varrho_0$ in which $\zeta = 0$. At larger

densities, $Y_p$ is always above the threshold of Urca processes, and $\zeta \neq 0$. At $T = 10$ MeV, the solid and dashed lines turn out to lie on top of one another.

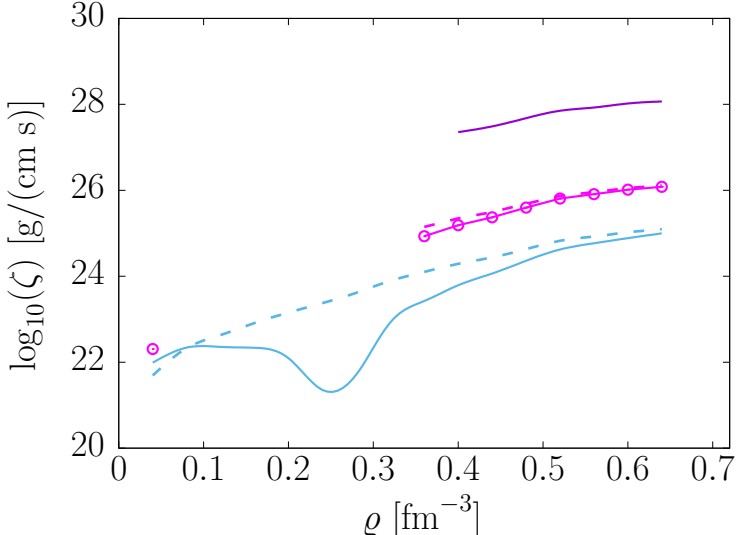

**Figure 5.** Density dependence of the bulk viscosity coefficient of $\beta$-stable matter associated with a density fluctuation of frequency $\omega = 2\pi \times 1$ kHz. The results were obtained by performing the derivatives of Equation (60) using both the isothermal (solid lines and open circles) and adiabatic (dashed lines) definitions. The labels specify the temperature in units of MeV.

It should be noted that, although the Urca process is always allowed at high temperatures, the corresponding values of $\zeta$ are very small, and the bulk viscosity is unlikely to be distinguished from other dissipative phenomena active in a neutron star merger.

The temperature dependence of the bulk viscosity coefficient at $\omega = 2\pi \times 1$ kHz and densities $\varrho = 2.5, 3,$ and $4 \varrho_0$ is illustrated in Figure 6. The maximum at $T \approx 2$ MeV, whose value monotonically increases with density, is clearly apparent. Owing to the low-temperature range spanned by the figure, in this instance the results obtained using isothermal and adiabatic derivatives turn out to be nearly identical. For this reason, only isothermal results are shown.

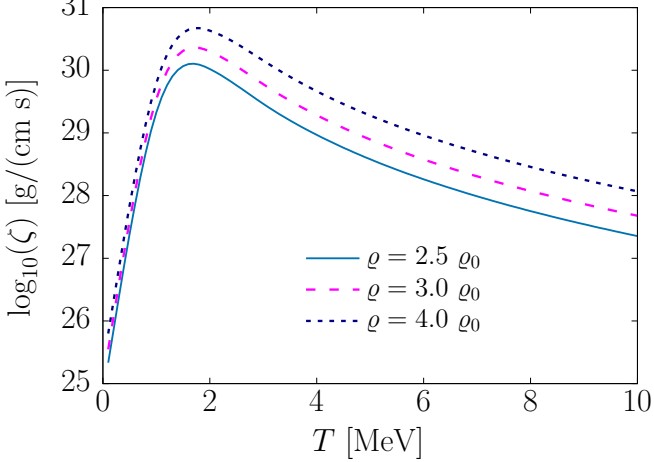

**Figure 6.** Temperature dependence of the bulk viscosity coefficient $\zeta$ corresponding to $\omega = 2\pi \times 1$ kHz and different densities.

## 7. Discussion

The description of neutron star mergers, which will be needed for the interpretation of current and future astronomical observations, requires a quantitative understanding of

both the equilibrium and dynamical properties of hot and dense matter. In this context, the availability of a dynamical model strongly constrained by phenomenology and suitable for use in finite-temperature perturbation theory will be crucial.

The approach described in this paper has been employed to obtain the EOS of matter with arbitrary neutron excess in the density region extending up to 4 $\varrho_0$—in which the applicability of the description in terms of nucleons is supported by electron-nucleus scattering data [50]—and temperatures up to 50 MeV. In this density regime—relevant to the description of neutron stars of mass compatible with the observational upper bound—the nonrelativistic approximation is expected to be justified, because the speed of sound in matter predicted by our approach never exceeds the speed of light.

Single-nucleon properties, such as the quasipartlcle spectra and effective masses, have also been computed within the same theoretical framework, in which thermal effects are consistently taken into account. Exploratory studies of dissipative processes leading to the damping of density oscillations in neutron star mergers suggest that a detailed treatment of thermal effects is, in fact, needed to clarify important issues, such as the onset of the Urca process.

The capability of consistently including muons is an important feature of our description of neutron star matter. However, in view of the fact that the most striking thermal effect emerging from the present study—that is, the breakdown of the monotonic density dependence of the proton fraction—is not significantly affected by the presence of muons (see Figure 3), the present analysis limited the discussion of bulk viscosity in the simpler case of *npe* matter. The role of muons in neutron star mergers is thoroughly discussed in Ref. [51].

The extension of the approach outlined in this paper to the description of neutrino emission processes and the neutrino mean free path along the line described in Refs. [24,25] does not involve any conceptual issues and is currently under way. In principle, the description of Urca processes involving hyperons [52] may also be possible. However, it would require a generalisation of the effective Hamiltonian allowing for a consistent description of all baryonic interactions. The development of such a model is beyond the scope of our work.

**Author Contributions:** All authors contributed to conceptualisation, development of the formalism and computing codes, and manuscript preparation. All authors have read and agreed to the published version of the manuscript.

**Funding:** This research was funded by the U.S. Department of Energy, Office of Science, Office of Nuclear Physics, under contract DE-AC02-06CH11357 (A.L.); the NUCLEI SciDAC program (A.L.); and the Italian National Institute for Nuclear Research (INFN), under grant TEONGRAV (O.B. and L.T.).

**Data Availability Statement:** The data supporting the conclusions of this study are available in the cited literature and within the present article.

**Conflicts of Interest:** The authors declare no conflict of interest.

## Note

1    In this article, we adopt the system of natural units, in which $\hbar = c = k_B = 1$, and, unless otherwise specified, neglect the small proton–neutron mass difference.

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
