# Peer review of "Properties of Hot Nuclear Matter"

_universe, doi:10.3390/universe9080345_

Round 1

Reviewer 1 Report

Referee report on "Properties of hot nuclear matter" by Benhar et al. submitted to Universe

The paper describes the correlated basis function (CBF) approach to properties of dense nuclear matter at finite temperature. 

It is essentially divided in two parts. The first part (sec. 2--5) reviews the CBF approach and describes an equilibrium properties of hot dense nuclear matter, while the second part (Sec. 6) is devoted to the calculation of the bulk viscosity.

The subject and a content of the manuscript by themselves are suitable for publication in the special issue Universe on Many-Body Theory. 

The first part is very well written, and I have only minor comments here (which are given at the end).

The bulk viscosity part, however, raises some doubts about the validity of the approach.

The bulk viscosity of the NS matter at finite temperatures was considered in several papers including the recent studies motivated by the NS-NS coalescence GW detection, see, for instance Alford, Harutyunyan, Sedrakian, Particles 5, 361-378 (2022) https://ui.adsabs.harvard.edu/abs/2022Parti...5..361A/abstract and references therein.

This detailed study is probably missed by the authors of the manuscript under review (since it is not cited). For instance, it describes a formalism of the calculation of the bulk viscosity coefficients for direct Urca reactions at finite temperatures for neutrino-trapped and neutrino-transparent regimes and for matter with or without muons.

In contrast, it is not clear how (what expressions were used for the reaction rates) the bulk viscosity is calculated in the present work. 

I emphasize several key points below.

1. At finite temperature there is no threshold for direct Urca process, which appears in Fig. 5 and discussed in Sec.6. There bulk viscosity should be non-zero in a whole density range of Fig. 5 for temperatures shown. It seems that the bulk viscosity is calculated in zero-temperature formalism, which of course is much simpler in terms of the phase-space integration but is incorrect (see the Alford et al. paper mentioned above and references therein).

2. In Sec.5, the importance of muons is emphasized. However, in sec. 6 muons silently disappear from the consideration. They are neither not present in Eqs. (52-53) nor discussed in Sec. 6. For instance, what beta-equilibrium fractions are used in Sec. 6? The ones calculated with muons or without them? Also the susceptibilities modify when the number of components in plasma increase (again, I refer here to the Alford et al. paper for simplicity).

3. It is assumed that the matter is transparent to neutrinos without any comments (lines 173-174). However, the typical temperature at which neutrinos are usually considered to be trapped in the NS interiors  is about 5 MeV (see, again, for instance Alford et al paper). Anyhow, the applicability of the neutrino-transparent approach needs to be discussed. 

Therefore, in my opinion, the manuscript cannot be published with Sec.6 in present from. The bulk viscosity should be recalculated using the proper finite-temperature formalism.

Minor comments

1. line 35. It is probably better to use \sim instead of \approx given that the pion mass is 134 -- 139 MeV depending on charge.

2. lines 104-105. Define S and T. 

3. line 105. The radial dependence of v^eff is shown in figure 1 for several baryon densities. It is unclear,  is it calculated for PNM or SNM or other proton fraction (this also applies for the whole sec. 2.2) as well. 

4. Eq. (22). It seems that N is missing in the last term (which includes \mu  -- cf. Eq. (21))

5. line 133. v -> V.

6. Eq. (25). Probably the terms containing \mu N_k are missing.

7. Particle fractions Y_k are used from the beginning of Sec.4, however are defined only at line 148.

8. First paragraph of Sec.5. It would be good to give a typical values of chemical potentials to give a reader a feeling on the magnitude of the ratio 2t/mu discussed here.

9. Eqs (52)-(53) repeat Eq. (38).

10. Below Eq.(59). Ellipses -> ellipsis (there is only one ellipsis there).

11. It would be more convenient to have the same dimensions of x-axes in figure 5 and figures 2-3. Especially since the authors refer to figure 3 when discuss figure 5.

12. Line 234. Is 1 kHz typical for pulsations in general or for oscillations in mergers? The reference here would help.

I do not have any significant comments on English. Some commas are missed after equations (e.g., after Eq. (5), after Eq. (31)). This should be checked once more.

Reviewer 2 Report

The manuscript presents the nuclear matter model based on the phenomenological nuclear Hamiltonian at finite and vanishing temperatures. It's shown how the inclusion of the two- and three-nucleon potentials, as well as the temperature effects, make the model to be applicable to the isolated NS and NS mergers. I consider the paper to be very well-written and interesting. I believe the comments presented below could improve the manuscript.

- The authors don't discuss the drawbacks/the applicability range of the approach. This should be clearly stated.

- In lines 173-174 it's written 'The results have been obtained assuming that neutrinos do not interact with matter, and have therefore vanishing chemical potential'. How it's correct at T=50 MeV? How it agrees with the fact that neutrinos are trapped in the star?

- In Eq. 41 it's mentioned that the parameter U_\alpha depends both on temperature and density. Could you elaborate more about it? What is the dependence and how it's determined?

- The authors have mentioned modification of the DU threshold with T. It would be interesting to see this change in the DU threshold on the figure.

- In the last sentence of the Conclusions it's mentioned about the future prospects related to account for the neutrino mean free path and neutrino emission processes. However, the inclusion of Lambda-hyperons at the considered densities is very important and should be taken into consideration.

Below there are some typos:

1) line 11 and further in the text, 'post merger' -> 'post-merger'

2) line 33, 'grand canonical potential'

3) In line 74, OPE potential isn't defined

4) line 96, one extra dot

5) page 7, between Eq. 34 and 35, 'thermodynamic consistence' -> 'thermodynamic consistency'

6) line 172, '\beta-equilibrium of matter' -> '\beta-equilibrium matter'

7) after line 205, 'is based of the' -> 'is based on the'

Round 2

Reviewer 1 Report

I am generally satisfied with the changes in the manuscript in response to my first report and think that the paper can now be published in Universe in present form.